# Impaired Performance of Broiler Chickens Fed Diets Naturally Contaminated with Moderate Levels of Deoxynivalenol

**DOI:** 10.3390/toxins13020170

**Published:** 2021-02-22

**Authors:** Regiane R. Santos, Ellen van Eerden

**Affiliations:** Department of Research and Development, Schothorst Feed Research, Meerkoetenweg 26, 8218 NA Lelystad, The Netherlands; EvEerden@schothorst.nl

**Keywords:** deoxynivalenol, Ross broilers, intestine, performance, carryover

## Abstract

Mycotoxin exposure is common in the poultry industry. Deoxynivalenol (DON) is usually detected at levels below the maximum threshold (5000 ppb), but depending on diet and age, broiler performance can be affected. We evaluated the effects of 900 ppb and 2300 ppb DON on the performance, intestinal morphometry, and lesion scores of broiler chickens. One-day-old male Ross broilers (*n* = 736) were divided into 4 treatments with 8 replicates each, and a pen containing 23 birds was the experimental unit. The animals were fed diets naturally contaminated with two levels of DON: 900 (Low DON—LD) or 2300 (Moderate DON—MD) ppb, with or without activated charcoal, over 28 days. After this, all birds were fed a marginally DON-contaminated diet without charcoal. During the first 28 days, body weight gain (BWG) and feed conversion ratio (FCR) were significantly impaired when broilers were fed a MD diet without activated charcoal. Even after feeding a marginally contaminated diet from D28–35, birds previously fed the MD diet presented a significantly lower performance. The villus height:crypt depth (VH:CD) ratio was significantly higher in the ileum from 14-day-old broilers fed the MD when compared with the LD diet. At D28, the MD diet caused decreased villus height (VH) and increased crypt depth (CD), affecting VH:CD ratio in both intestinal segments, with higher levels in the jejunum from 28-day-old broilers fed a non-supplemented LD diet. Broiler production was negatively affected by DON, even at moderate levels (2300 ppb).

## 1. Introduction

Among the factors that negatively affect animal production, mycotoxins play an important role because these natural contaminants are ubiquitous contaminants in feeds and feedstuffs [1]. Besides this, cereal byproducts that are rejected for human consumption during processing for mycotoxin removal, as well as raw materials such as dried distillers’ grains with solubles (DDGS), can enter the animal feed chain, increasing the exposure risk [2]. In most cases, contamination levels are low enough to ensure compliance with feed safety recommendations. However, mycotoxin contamination might still exert adverse effects on animals, and, at an economic level, the major mycotoxins risks are linked to suboptimal production and not to disease.

There is evidence that even at levels below authorities’ feed safety recommendations, deoxynivalenol (DON) may decrease resistance to infectious disease in broilers [3]. A previous study showed that broilers fed a diet containing 3000–4000 ppb DON, i.e., at levels below the European maximum guidance (5000 ppb in the complete diet) [4], are predisposed to develop necrotic enteritis. Also, broilers fed 1500 ppb DON combined with 20,000 ppb fumonisins are more susceptible to coccidiosis [5]. This poses a concern for the broiler industry, and there is an ongoing discussion about the reduction of antibiotics and the potential impact on the use of anticoccidials, particularly those from the class of ionophores. This class of anticoccidial fits the classical definition of an antibiotic because they have some antibacterial activity. This means that the importance of mycotoxins in the poultry industry may increase in a situation where ionophore anticoccidials are banned from feed. Besides this, the subclinical and indirect effects of mycotoxins are often underestimated because no typical mycotoxicosis symptoms are observed. In general, mycotoxins may be involved in numerous subclinical symptoms, and will potentiate the negative effect of diseases or simply lead to impaired performance.

To mimic on-farm conditions, it is important to expose broilers to feed naturally contaminated with mycotoxins as opposed to experimental contamination with synthetic mycotoxins. Furthermore, realistic contamination levels have to be considered when applying naturally contaminated diets. For instance, it is not common to find feedstuffs highly contaminated with fumonisins. Therefore, the probability of producing a diet with a final concentration between 15,000 ppb and 20,000 ppb (20,000 ppb is the European threshold level in complete feed) [4] is extremely low. The maximum acceptable level of DON in cereals and cereal products used for feed production is 8000 ppb, while for maize byproduct feed materials, it is 12,000 ppb (EC 576/2006) [4]. Considering that northwestern European broiler diets contain approximately 30% wheat, it will be difficult to reach 4000 ppb in the final diet. Based on this, we can expect only low to moderate levels of DON in broiler diets, e.g., 1000 ppb to 3000 ppb.

The type of feedstuff added to the diet also interferes with intestinal health and broiler performance. Wheat and rye evoke increased intestinal viscosity in broilers due to high levels of soluble non-starch polysaccharides (NSP), and this results in impaired nutrient digestibility and predisposes to infections [6,7]. This effect is usually prevented by the inclusion of enzymes that break down soluble NSP. For this trial, we decided to keep the inclusion level of wheat close to practice, whereas a small amount of rye was included to induce a mild intestinal challenge.

For this study, we exposed broilers during the starter and grower phase to two different levels of DON (900 ppb or 2300 ppb). In the finisher period, they received a diet with a negligible level of DON (57.3 ppb) to evaluate carryover effects. As a positive control, activated charcoal was tested at each DON level. Furthermore, feed was not supplemented with coccidiostats or NSP enzymes, with the aim to evaluate animal performance and intestinal integrity.

## 2. Results

### 2.1. Broiler Performance

The average body weight of the birds at the start of the trial was 42.7 g (42.4–43.0 g) for all treatments. The lowest body weight (BW) at D14 and D28 was observed in broilers fed the moderate DON (MD), regardless of the dietary supplementation with activated charcoal. Dietary supplementation with activated charcoal improved BW only at D14. Although birds were fed a diet with negligible levels of DON and no activated charcoal at D35, the birds previously fed with MD diets presented a significantly lower BW (Table 1).

During the starter (D0–14; Table 2) and grower (D14–28; Table 3) periods, there were no interactions between DON level and activated charcoal. The lowest body weight gain (BWG) and highest feed conversion ratio (FCR) at D14 and D28 was observed when birds were fed the MD diet, regardless of the dietary supplementation with activated charcoal (Table 2 and Table 3). Activated charcoal improved BWG at D14 only but did not improve FCR (Table 2). None of the diets affected feed intake (FI) in the starter (D0–14) and grower (D14–28) periods (Table 2 and Table 3).

During the finisher phase (D28–35), the broilers were fed a marginally contaminated diet (57.3 ppb DON) without activated charcoal. No dietary effects were observed in the performance parameters in this period (Table 4).

Considering the complete feeding period (D0–35; Table 5), no interactions between DON and activated charcoal were observed, but broiler chickens fed the MD diet during the starter and grower phase presented a significantly lower BWG and a higher FCR, even if they were fed a marginally contaminated diet in the finisher period.

### 2.2. Intestinal Analysis

#### 2.2.1. Jejunum and Ileum Morphometry

The effects of DON and activated charcoal on jejunum and ileum morphometry were evaluated by comparing villus height (VH) (µm), crypt depth (CD) (µm), villus height:crypt depth (VH:CD) ratio, and villus area (µm^2^) among the treatments at D14 (Table 6), D28 (Table 7), and D35 (Table 8). The VH:CD ratio was significantly higher in the ileum from 14-day-old broilers fed the MD when compared with the LD diet (Table 6). The VH and VH:CD ratio in the jejunum from 28-day-old broilers showed a significant interaction, indicating that broilers fed a non-supplemented MD diet had the shortest villi and lowest VH:CD ratio when compared with the non-supplemented LD diet. These differences were statistically significant. When LD diets were supplemented with activated charcoal, a significant decrease in VH was observed (Table 7).

Regarding the ileum, the crypt depth was highest when broilers chickens were fed a non-supplemented MD diet and lowest when the birds were fed the non-supplemented LD diet. This resulted also in the significantly lowest and highest VH:CD ratio when the birds were fed the MD and LD diets, respectively. No dietary effect was observed on the morphometry of the jejunum and ileum from 35-day-old broilers (Table 8).

#### 2.2.2. Goblet Cell Counting

The effects of DON and activated charcoal on jejunum and ileum mucus production were evaluated by counting the number of goblet cells per villus and by determining the density of these goblet cells according to the villus area (µm^2^) among the treatments at D14 (Table 9), D28 (Table 10), and D35 (Table 11). Dietary effects were observed only at D14, where a significant increase in goblet cell density was observed in the ileum from broilers fed the MD diet, regardless of the supplementation with activated charcoal (Table 9).

#### 2.2.3. Intestinal Lesion Scores

Lesion scores were given to the jejunum and ileum on a scale from 0 to 6 based on the Chiu/Park scoring method, where the higher the score, the higher the degree of damage. At D14, lesion scores were significantly increased when birds were fed a MD diet supplemented with activated charcoal, whereas at D28, the highest lesion scores were observed in broiler chickens receiving a non-supplemented MD died. No dietary effects were observed at D35 (Table 12). Figure 1 illustrates examples of jejunum and ileum sections from broilers fed MD and LD diets supplemented or not with activated charcoal.

#### 2.2.4. Intestinal Viscosity

The results concerning viscosity in the duodenum on D14, D28, and D35 are shown in Table 13. Intestinal viscosity was significantly increased in 14-day-old broilers fed the MD diet, regardless of the dietary supplementation with activated charcoal. No other changes were observed in the other feeding phases.

## 3. Discussion

In the present study, we demonstrate that even at moderate levels (2300 ppb), DON can impair the performance of broiler chickens. Also, this negative effect is not mitigated when the chickens are subsequently fed a diet with negligible levels (57.3 ppb) of DON for 7 days. Anticipating the potentially increasing impact of mycotoxins when ionophore anticoccidials are banned from feed, we decided not to supplement the experimental diets with anticoccidials. In a previous study, broiler chickens fed diets containing 1600 ppb DON and simultaneously challenged with *Eimeria* spp. [5] had an impaired performance, and one of the factors involved was the ability of DON to modulate the host immune response to coccidial infections [8]. In the present study, instead of challenging the birds with *Eimeria* spp. oocysts, we excluded anticoccidials from the diet formulation. Coccidiosis not only leads to clinical signs but can also result in poor performance [9]. Another factor that can challenge gut function is an increase in digesta viscosity, because it can decrease nutrient digestibility [7] and increases the retention time of the diet in the intestine, inducing competition with gut microbiota for digestible nutrients. Subsequently, this increases the risks of infection [10]. Therefore, in the present study, we used a wheat-based diet without NSP enzymes together with rye at an inclusion level of 5% in the starter (D0–14) and grower (D14–28) diets, thus increasing viscosity.

Although the diets were contaminated with a variety of mycotoxins, the negative impact observed in the present study was basically caused by DON. Besides DON, the other mycotoxins present in the diets were deoxynivalenol-3-glucoside (DON-3-G), enniatins B and B1 (ENNB+B1), zearalenone (ZEN), ochratoxin A (OTA), alternariol (AOH), and alternariol methyl ether (AME). The levels of ZEN, OTA, AOH, and AME were negligible. When DON is conjugated with glucose, resulting in the modified form of DON-3-G, it becomes unable to bind to the ribosome peptidyl transferase center, the main target of intestinal toxicity [11]. In pigs but not in broiler chickens, DON-3-G can be hydrolyzed to DON [12]. Therefore, the observed dietary levels between 10.7 ppb and 1670 ppb in the present study should not be a reason for concern. Enniatin B has a low intestinal toxicity in broiler chickens [13]. Furthermore, it was remarkable that ENNB+B1 levels were higher in MD than in LD diets.

Increasing the DON level from 900 ppb to 2300 ppb in the starter and grower diets resulted in a decreased BWG. This negative effect on BWG was partly counteracted by activated charcoal only in the MD diet. The same pattern was observed with FCR, which was negatively affected in broilers fed the MD diet without activated charcoal. One must bear in mind that the goal of this study was not to promote activated charcoal as a DON adsorbent, since it may also absorb nutrients [14], but to use it as an extra control [15]. Instead, it was shown that even when supplementing a diet with a compound that can bind DON, performance losses are not completely avoided. When the diet was replaced by a marginally contaminated feed in the finisher phase (D28–35), no differences in performance were observed in this period. However, the final body weight of the birds fed the MD diet during the starter and grower period was impaired, regardless of the supplementation with activated charcoal. Moreover, considering the complete production period (D0–35), BWG and FCR were impaired in broilers fed the MD diet compared with those fed the LD diet, whereas FI was not affected by the experimental diets. Based on a previous study, DON has a low oral availability (~19%) and a high plasma clearance (~0.12 L/min kg) in broilers [16]. Therefore, the impaired performance was not a result of mycotoxin accumulation or anorexia caused by DON. Previously, Lucke et al. [17] did not observe any difference in broiler performance after 28 days of dietary exposure to 5000 ppb DON, and only observed differences after 35 days when BWG was significantly decreased. Besides the dietary differences, in this later study, broiler chickens were fed diets artificially contaminated with DON. Importantly, to achieve homogeneity, this mycotoxin was mixed with inulin at a rate of 0.03% to 0.20% depending on the desired DON dietary level. Inulin is a fructo-oligosaccharide (FOS) able to increase the production of cecal butyric acid, which provides energy for the growth of the intestinal epithelium [18]. In another study, broiler performance was even improved, and intestinal viscosity decreased when the diet was contaminated with 1500 ppb DON [19]. The former authors suggested that physicochemical alterations in the wheat, caused by mycotoxins, resulted in better nutrient digestibility and in a lower viscosity in the jejunum and ileum. However, we did not observe a similar decrease in viscosity, which was measured in the duodenum. Instead, we observed an increase when birds were fed the MD diet during the starter phase. The prolonged exposure time to DON did not affect duodenal viscosity at D28 (8–9 cP), although it was still considerably higher than that observed in birds fed a marginally contaminated diet in the finisher period (2.2 cP). Also, the finisher diet contained NSP enzymes, which can explain the reduced viscosity in this phase. In a similar study, Dänicke et al. [19] fed a control diet resulting in high ileal viscosity (above 25 mPa-s, i.e., 25 cP), which was considerably higher than that observed in our study. Viscosity can be increased by including rye and excluding NSP enzymes [20] as we did in the present study. An increase in viscosity will negatively influence nutrient digestibility and absorption, decreasing BWG and increasing FCR. Therefore, it is not surprising that this diet, combined with moderate DON contamination, impaired broiler performance. Nonetheless, differences in DON levels and dietary composition do not allow a comparison between the study of Dänicke et al. [19] and the current study. Also, the wheat batch we used was naturally contaminated with a mixture of mycotoxins and not submitted to induced infection with a specific *Fusarium* strain, and nutritional composition may vary among batches. Recently, we showed that broiler chickens fed a corn-based diet (low in NSP) contaminated with ~3500 ppb DON and ~50 ppb of its derivatives 3+15 Ac-DON presented an impaired FCR, indicating that intestinal viscosity is not crucial to observe the negative impact of DON on performance [21]. Likewise, a longitudinal study also showed that mixtures of mycotoxins below EU recommendation levels impair the performance of broiler chickens [22].

Intestinal morphometric changes in the first 14 days of exposure were negligible and limited to an increase in the VH:CD ratio in the ileum of broilers fed the MD diet, regardless of the presence of activated charcoal, indicating that cell proliferation decreased in broilers fed the MD diet without resulting in immediate villus shortening. This was expected, since DON decreases cell proliferation and the crypt is responsible for cell renewing and maintenance of villus length. After 28 days, however, broilers fed the MD diet presented a significantly lower villus height and VH:CD ratio than those fed the LD diet. This leads us to infer that birds fed the MD diet were probably trying to maintain villus height by a compensatory increase in the proliferation in the crypt, which required extra energy. Besides this, shortened villi and deeper crypts will result in suboptimal nutrient absorption and impaired animal performance [23,24]. This fits with the observed decreased BWG and increased FCR at D28. Besides, the ileum presented a higher VH:CD ratio due to an increase in crypt depth, showing that in this intestinal section, cell proliferation was increased to keep villus height similar to the expected (LD diet) levels. Such a compensatory mechanism costs energy and will result in suboptimal performance. At D35, no differences were observed, because the finisher diet had negligible levels of DON and intestinal cell turnover takes around 48 h to 96 h [25]. The number of goblet cells was not affected by the treatments. In a study with pigs, it was shown that DON causes oedema in the *lamina propria* and contact loss between *lamina propria* and enterocytes [26]. However, in the present study, the tested DON levels were not able to cause such effects. Based on the morphometric analysis and lesion scores in the jejunum and ileum, it can be confirmed that the jejunum was the intestinal section more sensitive to DON than the ileum, as reported before for broilers submitted to DON exposure [27] or other sources of stress [28].

In conclusion, broiler chickens fed a diet containing moderate levels of DON (2300 ppb) will perform inefficiently. Further, the influence of mycotoxins on poultry performance should be assessed not only on its toxicity per se, but also considering animal age, dietary composition, and the presence of other types of additives, such as NSP enzymes and anticoccidials.

## 4. Materials and Methods

### 4.1. Animal Ethics Statement

The experiment was conducted according to the guidelines of the Animal and Human Welfare Codes/Laboratory practice codes in the Netherlands. The protocol was approved by the Ethics Review Committee: Body of Animal Welfare at SFR (AVD246002016450), approval date: 28 February 2019.

### 4.2. Broilers and Housing

One-day-old male Ross 308 broilers purchased from a local commercial hatchery were used in this study, with 4 dietary treatments of 184 chicks each (divided among 8 replicate pens with 23 chicks each). The birds were housed in 32 floor pens with wood shavings as bedding material in the broiler facilities of Schothorst Feed Research, Lelystad, The Netherlands. Each pen (2.2 m^2^) had 1 feeder and 3 drinking nipples. Birds were kept until 35 days of age. The ambient temperature was gradually decreased from 34.5 °C at the arrival of the birds to 19.4 °C at 35 days of age. Room temperature and relative humidity were recorded daily. Light was provided continuously for the first 24 h to give birds the opportunity to readily find feed and water. After that, the light schedule was 22L (light): 2D (dark) for 1 day and then 8L: 4D: 10L: 2D for the remaining experimental period, complying with European Union (EU) legislation of a minimum of 6 h of darkness from the second day onward, of which at least 4 h was uninterrupted darkness. Birds were vaccinated against Newcastle Disease at D10 and against Infectious Bursal Disease at D20 of the trial. The health status of the flock was monitored by a poultry veterinarian. Throughout the experimental period, all animals were monitored daily for abnormalities, such as abnormal behavior, clinical signs of illness, and mortality.

### 4.3. Diets and Experimental Design

The experiment comprised 4 dietary treatments in a factorial design with DON (2 levels; moderate—MD and low—LD) in diets with or without activated charcoal (2 g/kg diet; Norit, Carbomix, KELA Pharma, Sint-Niklaas, Belgium), applied as a DON adsorbent [15]. Diets were prepared with wheat batches naturally contaminated with different DON levels (1650 ppb and 6880 ppb), together with other mycotoxins. The recommended maximum level of DON in poultry diet is 5000 ppb (EU Commission Directive 2003/100/EC). Therefore, in the present study, the obtained 900 ppb and 2300 ppb DON in the final diet were considered as low (LD) and moderate (MD) DON levels, respectively. Treatments were randomly allocated per block to pens, where each treatment was repeated 8 times. The pen was the experimental unit, and each pen contained 23 broilers. Dietary treatments are summarized in Table 14, together with their mycotoxin composition. Although the supplemented diets were made from the same basal diet of each MD or LD level, the levels of DON were measured in all diets to calculate the mean contamination level. The mean levels of DON in the LD diets during the starter and grower phases were 881 ± 4 ppb and 876 ± 92 ppb, respectively. The mean levels of DON in the MD diets during the starter and grower phases were 2130 ± 99 ppb and 2290 ± 99 ppb, respectively. The finisher diet was prepared with marginally contaminated feedstuffs, reaching a DON level of 57 ppb. All diets were analyzed by an independent and accredited laboratory (Primoris, Belgium). A multi-mycotoxin test was applied, showing that DON was the main contaminant and that other mycotoxins were found at negligible levels. The nutrient composition of the diets is given in the Appendix A.

### 4.4. Performance and Litter Score

Broilers were weighed per pen on D0, D14, D28, and D35, and feed consumption and mortality were recorded throughout the experimental period. Body weight gain (BWG), feed intake (FI), and feed conversion ratio (FCR) were determined in the cumulative phases from D0–14, D14–28, D28–35, and D0–35. Litter quality was visually scored at D14, D28, and D35 on a scale of 1–10, with 1 indicating low quality (wet) and 10 indicating high quality (dry and friable) [29].

### 4.5. Intestinal Analysis

#### 4.5.1. Jejunum and Ileum Morphometry and Goblet Cell Counting

Samples of jejunum and ileum (we randomly selected 1 bird/pen on D14, D28, and D35) were collected and fixed in buffered formalin for histological analysis. In brief, histological slides (periodic acid–Schiff (PAS) counterstaining with hematoxylin staining) from the jejunum and ileum from each bird were scanned by the NanoZoomer-XR (Hamamatsu Photonics KK, Hamamatsu, Japan). The scanned slides were viewed through the viewer software (NDP.view2; Hamamatsu, Japan) and analyzed using the analysis software (NDP.analyze; Hamamatsu, Japan). Villus height (VH), crypt depth (CD), and villus area (µm^2^) from each individual bird were measured (5 villi per intestinal segment). The measurements of VH and CD were used to calculate the VH:CD ratio. The number of goblet cells and goblet cell density per villus were also quantified in scans of Alcian Bleu sections, serial to the PAS-hematoxylin sections. Only intact villi were measured.

#### 4.5.2. Jejunum and Ileum Lesion Scores

To evaluate the degree of mucosal damage, the Chiu/Park scale was applied [30]. In brief, the mucosa was classified from normal if presenting an intact structure with no visible damage (degree 0) to severely damaged (degree 6), as previously described [28]:Degree 0: Intact without visible damage.Degree 1: Damage in subepithelial space at villus tips.Degree 2: Extension of subepithelial space with moderate lifting.Degree 3: Massive lifting down the sides of villi with some denuded villi.Degree 4: Denuded villi with dilated capillaries.Degree 5: Disintegration of lamina propria.Degree 6: Crypt injury.

To statistically compare the degrees among the treatments, a composite score per treatment was determined by averaging the score from each bird. For this, the percentage of villi with a specific degree was multiplied by its respective degree. This calculation was performed for each degree per treatment, and the sum obtained was considered the composite score [31].

#### 4.5.3. Intestinal Viscosity

Digesta samples from the duodenum were collected at D14 (pooled sample of 3 birds per pen), D28 (1 bird per pen), and D35 (1 bird per pen). The samples were submitted to viscosity determination according to the AOAC protocol. In brief, each sample was centrifuged for 10 min at 3500× g and at 4 °C. The supernatant was filtrated with a serum filter tube and placed on ice. Viscosity of the supernatant was measured at 20 °C with a digital Brookfield DV-II LVCP viscometer (Brookfield Engineering, Middleboro, MA, USA) according to the “plate and cone” method.

### 4.6. Statistical Analysis

Observations were marked as outliers to be excluded from the dataset prior to statistical analyses if the residual (fitted—observed value) was more than 2.5× standard error of the parameter. If at least 1 of the response parameters FI, BWG, or FCR was an outlier, then all 3 records were dropped for that particular observation in that measurement period. Body weight was analyzed separately from the other production parameters. Data regarding intestinal morphometry and goblet cells counting were used without removing outliers. The experimental data were analyzed with ANOVA (GenStat Version 19.0, 2018, Hemel Hempstead, UK). Each pen was an experimental unit. Given the factorial design, the statistical model used to analyze the data was:
Y = μ + block_i_ + DON_j_ + Additive_k_ + DON∗Additive_jk_ + e_ijk_

In which:
Y = Response parameterμ = General meanBlock_j_ = Effect of block (i = 1…8)DONj = Effect of DON (j = 1, 2)Additivek = Effect of Additive (k = 1, 2)DON*Additivejk = Effect of the interactions between DON and AdditiveErrorijk = Error term

Treatment means were compared by least significant difference (LSD). Values with *p* ≤ 0.05 were considered statistically significant.

## Figures and Tables

**Figure 1 toxins-13-00170-f001:**
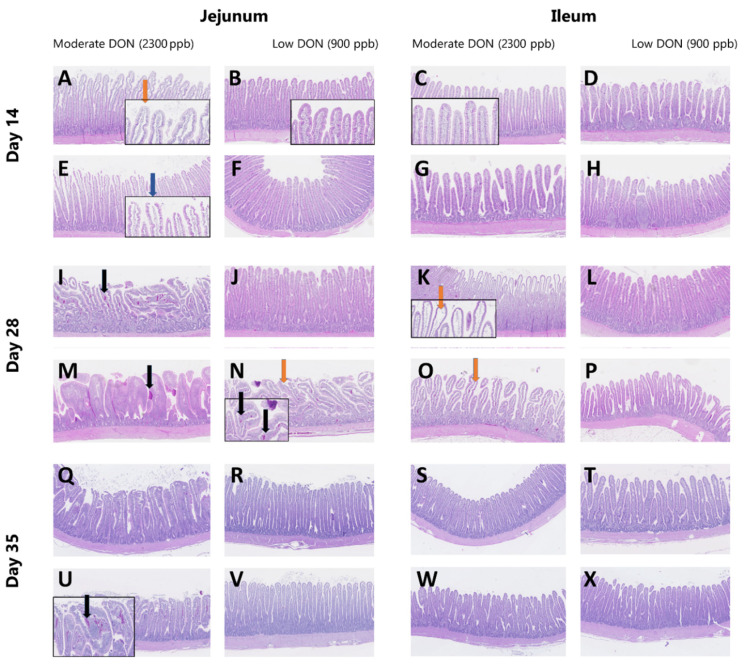
Illustrative images of periodic acid–Schiff (PAS)-hematoxylin-stained sections of jejunum and ileum from broilers fed the experimental diets. Orange arrows indicate denuded lamina propria, blue arrow shows damage in villus tip, and black arrows indicate blood in the villi. Scale bars = 100 μm.

**Table 1 toxins-13-00170-t001:** Mean (± SD) effects of deoxynivalenol (DON) and activated charcoal on body weight (BW; g) of the birds at D14, D28, and D35.

Treatment	DON	D14		D28		D35	
No Additive	MD	530 ± 9		1696 ± 69		2362 ± 88	
Activated Charcoal	MD	545 ± 20		1762 ± 117		2382 ± 89	
No Additive	LD	551 ± 23		1830 ± 52		2464 ± 71	
Activated Charcoal	LD	563 ± 11		1822 ± 78		2438 ± 95	
	MD	538 ± 17	a	1729 ± 99	a	2372 ± 86	a
	LD	557 ± 19	b	1826 ± 64	b	2451 ± 82	b
No Additive		541 ± 20	a	1763 ± 91		2413 ± 93	
Activated Charcoal		554 ± 18	b	1792 ± 101		2410 ± 94	
Effect		*p*-value	LSD	*p*-value	LSD	*p*-value	LSD
DON		0.004	12.5	0.001	54.8	0.02	64.6
Additive		0.042	12.5	0.28	54.8	0.94	64.6
DON × Additive		0.83	17.7	0.18	77.4	0.48	91.3

MD: Moderate DON level (2300 ppb in the diet); LD: Low DON level (900 ppb in the diet), LSD: Least significant difference. (a–c) Values followed by a different letter within a column differ significantly (*p* ≤ 0.05).

**Table 2 toxins-13-00170-t002:** Mean (± SD) effects of DON and activated charcoal in the starter phase (D0–14) on body weight gain (BWG; g), feed intake (FI; g), and feed conversion ratio (FCR; g/g) of broilers.

		BWG		FI		FCR	
Treatment	DON	(g)		(g)		(g/g)	
No Additive	MD	488 ± 9		578 ± 11		1.186 ± 0.016	
Activated Charcoal	MD	502 ± 20		586 ± 22		1.168 ± 0.011	
No Additive	LD	508 ± 23		588 ± 21		1.158 ± 0.020	
Activated Charcoal	LD	520 ± 11		597 ± 18		1.148 ± 0.035	
	MD	495 ± 17	a	582 ± 17		1.177 ± 0.016	b
	LD	514 ± 18	b	593 ± 20		1.153 ± 0.028	a
No Additive		498 ± 20	a	583 ± 17		1.172 ± 0.023	
Activated Charcoal		511 ± 18	b	592 ± 20		1.158 ± 0.027	
Effect		*p*-value	LSD	*p*-value	LSD	*p*-value	LSD
DON		<0.01	12.5	0.17	14.9	<0.01	0.0163
Additive		0.03	12.5	0.27	14.9	0.10	0.0163
DON × Additive		0.72	17.7	0.97	21.0	0.62	0.0231

MD: Moderate DON level (2300 ppb in the diet); LD: Low DON level (900 ppb in the diet). (a,b) Values followed by a different letter within a column differ significantly (*p* ≤ 0.05).

**Table 3 toxins-13-00170-t003:** Mean (± SD) effects of DON and activated charcoal in the grower phase (D14–28) on body weight gain (BWG; g), feed intake (FI; g), and feed conversion ratio (FCR; g/g) of broilers.

		BWG		FI		FCR	
Treatment	DON	(g)		(g)		(g/g)	
No Additive	MD	1166 ± 75		1814 ± 54		1.561 ± 0.080	
Activated Charcoal	MD	1217 ± 102		1849 ± 95		1.523 ± 0.078	
No additive	LD	1279 ± 38		1888 ± 56		1.476 ± 0.021	
Activated Charcoal	LD	1259 ± 71		1861 ± 70		1.480 ± 0.039	
	MD	1191 ± 91	a	1831 ± 77		1.542 ± 0.079	b
	LD	1269 ± 56	b	1875 ± 63		1.478 ± 0.030	a
No Additive		1222 ± 82		1851 ± 66		1.519 ± 0.072	
Activated Charcoal		1238 ± 88		1855 ± 81		1.501 ± 0.064	
Effect		*p*-value	LSD	*p*-value	LSD	*p*-value	LSD
DON		<0.01	50.4	0.10	51.2	<0.01	0.0401
Additive		0.52	50.4	0.88	51.2	0.39	0.0401
DON × Additive		0.16	71.3	0.23	72.4	0.31	0.0567

MD: Moderate DON level (2300 ppb in the diet); LD: Low DON level (900 ppb in the diet). (a,b) Values followed by a different letter within a column differ significantly (*p* ≤ 0.05).

**Table 4 toxins-13-00170-t004:** Mean (± SD) effects of DON and activated charcoal in the finisher phase (D28–35) on body weight gain (BWG; g), feed intake (FI; g), and feed conversion ratio (FCR; g/g) of broilers.

		BWG		FI		FCR	
Treatment	DON	(g)		(g)		(g/g)	
No Additive	MD	666 ± 105		1162 ± 81		1.767 ± 0.168	
Activated Charcoal	MD	620 ± 57		1148 ± 40		1.860 ± 0.124	
No Additive	LD	634 ± 42		1173 ± 62		1.854 ± 0.085	
Activated Charcoal	LD	616 ± 56		1141 ± 61		1.858 ± 0.078	
	MD	643 ± 85		1157 ± 62		1.814 ± 0.151	
	LD	625 ± 49		1149 ± 61		1.856 ± 0.079	
No Additive		650 ± 79		1168 ± 70		1.811 ± 0.136	
Activated Charcoal		618 ± 55		1144 ± 50		1.859 ± 0.100	
Effect		*p*-value	LSD	*p*-value	LSD	*p*-value	LSD
DON		0.49	52.7	0.92	45.5	0.36	0.0919
Additive		0.23	52.7	0.31	45.5	0.29	0.0919
DON × Additive		0.58	74.5	0.71	64.3	0.33	0.1299

MD: Moderate DON level (2300 ppb in the diet); LD: Low DON level (900 ppb in the diet). MD and LD diets were given in the start (D0–14) and grower (D14–28) periods only. During the finisher period, all birds were fed a diet with negligible (57.3 ppb) DON levels.

**Table 5 toxins-13-00170-t005:** Mean (± SD) effects of DON and activated charcoal in the complete feeding period (D0–35) on body weight gain (BWG; g), feed intake (FI; g), and feed conversion ratio (FCR; g/g) of broilers.

		BWG		FI		FCR	
Treatment	DON	(g)		(g)		(g/g)	
No Additive	MD	2319 ± 88		3555 ± 77		1.534 ± 0.041	
Activated Charcoal	MD	2339 ± 89		3583 ± 110		1.532 ± 0.038	
No Additive	LD	2421 ± 71		3649 ± 110		1.508 ± 0.026	
Activated Charcoal	LD	2396 ± 95		3600 ± 97		1.503 ± 0.028	
	MD	2329 ± 86	a	3569 ± 93		1.533 ± 0.038	b
	LD	2408 ± 82	b	3624 ± 104		1.505 ± 0.026	a
No Additive		2370 ± 93		3612 ± 104		1.521 ± 0.035	
Activated Charcoal		2368 ± 94		3591 ± 101		1.518 ± 0.036	
Effect		*p*-value	LSD	*p*-value	LSD	*p*-value	LSD
DON		0.02	64.5	0.13	73.2	0.04	0.0249
Additive		0.94	64.5	0.76	73.2	0.81	0.0249
DON × Additive		0.48	91.3	0.29	103.5	0.91	0.0352

MD: Moderate DON level (2300 ppb in the diet); LD: Low DON level (900 ppb in the diet). (a,b) Values followed by a different letter within a column differ significantly (*p* ≤ 0.05).

**Table 6 toxins-13-00170-t006:** Mean (± SD) effects of DON and activated charcoal on morphometric parameters and ileum at D14.

		Jejunum								Ileum							
		Villus Height		Crypt Depth		VH:CD	Villus Area			Villus Height		Crypt Depth		VH:CD		Villus Area	
Treatment	DON	(µm)		(µm)		ratio	(mm^2^)			(µm)		(µm)		ratio		(mm^2^)	
No Additive	MD	829 ± 110		160 ± 31		5.3 ± 0.6		81 ± 15		663 ± 44		158 ± 16		4.3 ± 0.5		64 ± 5.8	
Activated Charcoal	MD	830 ± 99		165 ± 31		5.2 ± 0.8		90 ± 17		671 ± 103		158 ± 47		4.6 ± 1.4		74 ± 29	
No Additive	LD	783 ± 72		180 ± 41		4.6 ± 1.4		82 ± 11		591 ± 69		195 ± 75		3.4 ± 1.0		63 ± 14	
Activated Charcoal	LD	841 ± 70		184 ± 33		4.7 ± 1.0		92 ± 21		670 ± 113		173 ± 21		4.0 ± 0.8		78 ± 20	
	MD	829 ± 101		162 ± 30		5.2 ± 0.7		86 ± 16		667 ± 77		158 ± 34		4.4 ± 1.0	b	69 ± 21	
	LD	812 ± 75		182 ± 36		4.7 ± 1.2		87 ± 17		630 ± 99		184 ± 54		3.7 ± 0.9	a	71 ± 18	
No Additive		806 ± 93		170 ± 37		5.0 ± 1.1		82 ± 13		627 ± 67		176 ± 56		3.8 ± 0.9		64 ± 11	
Activated Charcoal		835 ± 83		175 ± 32		5.0 ± 0.9		91 ± 18		671 ± 105		165 ± 36		4.3 ± 1.1		76 ± 24	
Effect		*p*-value	LSD	*p*-value	LSD	*p*-value	LSD	*p*-value	LSD	*p*-value	LSD	*p*-value	LSD	*p*-value	LSD	*p*-value	LSD
DON		0.59	66.1	0.13	25.6	0.14	0.75	0.83	13.2	0.25	63.8	0.07	28.0	0.03	0.66	0.85	13.7
Additive		0.37	66.1	0.71	25.6	0.97	0.75	0.16	13.2	0.17	63.8	0.43	28.0	0.18	0.66	0.08	13.7
DON × Additive		0.37	93.5	0.96	36.1	0.82	1.06	0.98	18.7	0.26	90.3	0.42	39.5	0.67	0.93	0.72	19.4

MD: Moderate DON level (2300 ppb in the diet); LD: Low DON level (900 ppb in the diet). (a,b) Values followed by a different letter within a column differ significantly (*p* ≤ 0.05).

**Table 7 toxins-13-00170-t007:** Mean (± SD) effects of DON and activated charcoal on morphometric parameters of jejunum and ileum at D28.

		Jejunum								Ileum							
		Villus Height		Crypt Depth		VH:CD		Villus Area		Villus Height		Crypt Depth		VH:CD		Villus Area	
Treatment	DON	(µm)		(µm)		ratio		(mm^2^)		(µm)		(µm)		ratio		(mm^2^)	
No Additive	MD	922 ± 113	a	232 ± 34		4.1 ± 0.9	a	182 ± 128		832 ± 192		235 ± 73	b	3.9 ± 1.4	a	119 ± 31	
Additive	MD	1002 ± 120	ab	214 ± 34		4.8 ± 0.9	ab	180 ± 168		876 ± 126		199 ± 41	ab	4.7 ± 1.4	ab	98 ± 19	
No Additive	LD	1073 ± 138	b	186 ± 23		5.9 ± 1.1	b	120 ± 16		951 ± 182		177 ± 26	a	5.5 ± 1.3	b	151 ± 113	
Additive Charcoal	LD	962 ± 153	a	212 ± 63		4.8 ± 1.2	ab	146 ± 63		888 ± 210		214 ± 41	ab	4.3 ± 1.2	ab	192 ± 171	
	MD	962 ± 120		223 ± 34		4.5 ± 1.0		181 ± 144		854 ± 158		217 ± 60		4.3 ± 1,4		109 ± 27	
	LD	1017 ± 152		199 ± 47		5.4 ± 1.2		133 ± 46		920 ± 193		196 ± 38		4.9 ± 1,4		171 ± 141	
No Additive		998 ± 144		209 ± 36		5.0 ± 1.3		151 ± 94		892 ± 191		206 ± 61		4.7 ± 1.5		135 ± 81	
Activated Charcoal		982 ± 134		213 ± 49		4.8 ± 1.1		163 ± 123		882 ± 167		206 ± 40		4.5 ± 1.3		145 ± 127	
Effect		*p*-value	LSD	*p*-value	LSD	*p*-value	LSD	*p*-value	LSD	*p*-value	LSD	*p*-value	LSD	*p*-value	LSD	*p*-value	LSD
DON		0.16	76.8	0.12	30.1	0.03	0.78	0.23	80.2	0.34	137.6	0.20	47.7	0.18	0.92	0.12	78.9
Additive		0.68	76.8	0.81	30.1	0.65	0.78	0.77	80.2	0.89	137.6	1.00	35.8	0.60	0.92	0.80	78.9
DON × Additive		0.02	108.6	0.15	42.6	0.03	1.10	0.72	113.5	0.43	194.6	0.04	37.0	0.041	1.305	0.43	111.6

MD: Moderate DON level (2300 ppb in the diet); LD: Low DON level (900 ppb in the diet). (a,b) Values followed by a different letter within a column differ significantly (*p* ≤ 0.05).

**Table 8 toxins-13-00170-t008:** Mean (± SD) effects of DON and activated charcoal on morphometric parameters of jejunum and ileum at D35.

		Jejunum								Ileum							
		Villus Height		Crypt Depth		VH:CD		Villus Area		Villus Height		Crypt Depth		VH:CD		Villus Area	
Treatment	DON	(µm)		(µm)		ratio		(mm^2^)		(µm)		(µm)		ratio		(mm^2^)	
No Additive	MD	1170 ± 169		256 ± 64		4.9 ± 1.3		212 ± 143		905 ± 124		205 ± 45		4.7 ± 1.6		170 ± 124	
Additive	MD	1166 ± 222		250 ± 61		4.9 ± 1.3		289 ± 156		887 ± 230		203 ± 33		4.5 ± 1.2		114 ± 39	
No Additive	LD	1120 ± 206		240 ± 55		4.9 ± 1.4		279 ± 126		835 ± 141		207 ± 65		4.3 ± 1.1		98 ± 24	
Additive Charcoal	LD	1175 ± 129		266 ± 43		4.6 ± 0.9		257 ± 155		895 ± 106		202 ± 36		4.6 ± 0.8		142 ± 80	
	MD	1168 ± 190		253 ± 60		4.9 ± 1.2		250 ± 150		896 ± 182		204 ± 38		4.6 ± 1.4		142 ± 90	
	LD	1148 ± 166		253 ± 49		4.7 ± 1.1		268 ± 138		865 ± 125		205 ± 50		4.4 ± 0.9		120 ± 62	
No Additive		1145 ± 182		248 ± 58		4.9 ± 1.3		245 ± 137		870 ± 133		206 ± 55		4.5 ± 1.3		134 ± 91	
Activated Charcoal		1171 ± 175		258 ± 51		4.7 ± 1.1		273 ± 151		891 ± 173		203 ± 33		4.5 ± 1.0		128 ± 63	
Effect		*p*-value	LSD	*p*-value	LSD	*p*-value	LSD	*p*-value	LSD	*p*-value	LSD	*p*-value	LSD	*p*-value	LSD	*p*-value	LSD
DON		0.78	145.4	1.00	43.8	0.72	0.98	0.74	105.6	0.60	120.7	0.99	36.2	0.72	0.87	0.39	51.5
Additive		0.72	145.4	0.65	43.8	0.81	0.98	0.60	105.6	0.72	120.7	0.86	36.2	0.99	0.87	0.83	51.5
DON × Additive		0.68	205.6	0.47	62.0	0.71	1.39	0.35	149.4	0.51	170.8	0.93	51.4	0.51	1.24	0.06	72.8

MD: Moderate DON level (2300 ppb in the diet); LD: Low DON level (900 ppb in the diet). MD and LD diets were given in the start (D0–14) and grower (D14–28) periods only. During the finisher period, all birds were fed a diet with negligible (57.3 ppb) DON levels.

**Table 9 toxins-13-00170-t009:** Mean (± SD) effects of DON and activated charcoal on the number and density of goblet cells in jejunum and ileum of broilers at D14.

Treatment	DON	Jejunum				Ileum			
		N° Goblet Cells		Goblet Cells/μm^2^		N° Goblet Cells		Goblet Cells/μm^2^	
No Additive	MD	111 ± 21		1.4 ± 0.2		104 ± 18		1.6 ± 0.2	
Activated Charcoal	MD	126 ± 29		1.4 ± 0.2		119 ± 63		1.6 ± 0.2	
No Additive	LD	99 ± 20		1.2 ± 0.3		84 ± 24		1.3 ± 0.3	
Activated Charcoal	LD	112 ± 29		1.2 ± 0.3		101 ± 41		1.3 ± 0.4	
	MD	119 ± 26		1.4 ± 0.2		111 ± 45		1.6 ± 0.2	b
	LD	106 ± 25		1.2 ± 0.3		93 ± 34		1.3 ± 0.3	a
No Additive		105 ± 21		1.3 ± 0.3		94 ± 23		1.5 ± 0.3	
Activated Charcoal		119 ± 29		1.3 ± 0.3		110 ± 52		1.4 ± 0.3	
Effect		*p*-value	LSD	*p*-value	LSD	*p*-value	LSD	*p*-value	LSD
DON		0.14	17.0	0.06	0.17	0.20	29.7	0.02	0.21
Additive		0.10	17.0	0.88	0.17	0.28	29.7	0.67	0.21
DON × Additive		0.90	24.0	0.92	0.24	0.91	42.0	0.86	0.29

MD: Moderate DON level (2300 ppb in the diet); LD: Low DON level (900 ppb in the diet). (a,b) Values followed by a different letter within a column differ significantly (*p* ≤ 0.05).

**Table 10 toxins-13-00170-t010:** Mean (± SD) effects of DON and activated charcoal on the number and density of goblet cells in jejunum and ileum of broilers at D28.

Treatment	DON	Jejunum				Ileum			
		N° Goblet Cells		Goblet Cells/μm^2^		N° Goblet Cells		Goblet Cells/μm^2^	
No Additive	MD	143 ± 56		1.0 ± 0.6		160 ± 88		1.4 ± 0.7	
Activated Charcoal	MD	151 ± 65		1.2 ± 0.7		134 ± 28		1.4 ± 0.4	
No Additive	LD	147 ± 41		1.3 ± 0.5		161 ± 49		1.4 ± 0.7	
Activated Charcoal	LD	164 ± 56		1.2 ± 0.5		172 ± 95		1.5 ± 1.1	
	MD	147 ± 59		1.1 ± 0.6		147 ± 65		1.4 ± 0.6	
	LD	155 ± 48		1.2 ± 0.5		167 ± 73		1.4 ± 0.9	
No Additive		145 ± 47		1.1 ± 0.5		160 ± 69		1.4 ± 0.7	
Activated Charcoal		157 ± 59		1.2 ± 0.6		153 ± 71		1.4 ± 0.8	
Effect		*p*-value	LSD	*p*-value	LSD	*p*-value	LSD	*p*-value	LSD
DON		0.64	36.5	0.52	0.40	0.41	47.2	0.95	0.50
Additive		0.49	36.5	0.67	0.40	0.76	47.2	0.87	0.50
DON × Additive		0.80	51.7	0.52	0.57	0.42	66.7	0.94	0.70

MD: Moderate DON level (2300 ppb in the diet); LD: Low DON level (900 ppb in the diet).

**Table 11 toxins-13-00170-t011:** Mean (± SD) effects of DON and activated charcoal on the number and density of goblet cells in jejunum and ileum of broilers at D35.

Treatment	DON	Jejunum				Ileum			
		N° Goblet Cells		Goblet Cells/μm^2^		N° Goblet Cells		Goblet Cells/μm^2^	
No Additive	MD	81.3 ± 36.5		0.5 ± 0.2		73.0 ± 26.5		0.6 ± 0.4	
Activated Charcoal	MD	65.0 ± 29.1		0.3 ± 0.2		72.2 ± 33.4		0.6 ± 0.1	
No Additive	LD	76.2 ± 46.7		0.4 ± 0.4		73.6 ± 21.8		0.8 ± 0.3	
Activated Charcoal	LD	84.8 ± 28.8		0.5 ± 0.3		65.5 ± 34.8		0.6 ± 0.3	
	MD	73.1 ± 33.0		0.4 ± 0.2		72.6 ± 30		0.6 ± 0.3	
	LD	80.5 ± 37.0		0.4 ± 0.3		69.5 ± 28		0.7 ± 0.3	
No Additive		78.7 ± 40.1		0.4 ± 0.3		73.3 ± 23		0.7 ± 0.3	
Activated Charcoal		74.9 ± 29.8		0.4 ± 0.3		68.8 ± 33		0.6 ± 0.2	
Effect		*p*-value	LSD	*p*-value	LSD	*p*-value	LSD	*p*-value	LSD
DON		0.59	27.9	0.89	0.21	0.73	17.8	0.38	0.16
Additive		0.78	27.9	0.63	0.21	0.61	17.8	0.22	0.16
DON × Additive		0.37	39.5	0.20	0.30	0.68	25.2	0.10	0.22

MD: Moderate DON level (2300 ppb in the diet); LD: Low DON level (900 ppb in the diet). MD and LD diets were given in the start (D0–14) and grower (D14–28) periods only. During the finisher period, all birds were fed a diet with negligible (57.3 ppb) DON levels.

**Table 12 toxins-13-00170-t012:** Mean (± SD) effects of DON and activated charcoal on jejunum and ileum Chiu/Park intestinal lesion scores *.

		Jejunum						Ileum					
Treatment	DON	D14		D28		D35		D14		D28		D35	
No Additive	MD	0.3 ± 0.2	a	3.1 ± 1.3	b	1.5 ± 0.7		0.1 ± 0.1		1.3 ± 0.8		0.7 ± 0.7	
Additive	MD	1.1 ± 0.8	b	2.2 ± 1.4	ab	1.2 ± 1.3		0.3 ± 0.3		1.2 ± 1.2		0.9 ± 0.9	
No Additive	LD	1.0 ± 1.2	ab	1.3 ± 1.6	a	1.0 ± 0.7		0.2 ± 0.2		0.6 ± 0.6		0.6 ± 0.6	
Additive Charcoal	LD	0.2 ± 0.4	a	2.3 ± 1.2	ab	0.3 ± 0.6		0.0 ± 0.0		1.5 ± 1.1		0.4 ± 0.3	
	MD	0.7 ± 0.7		2.7 ± 1.4		1.3 ± 1.0		0.2 ± 0.2		1.2 ± 0.9		0.8 ± 0.8	
	LD	0.6 ± 0.6		1.8 ± 1.5		0.6 ± 0.7		0.1 ± 0.1		1.0 ± 1.0		0.5 ± 0.5	
No Additive		0.7 ± 0.7		2.2 ± 1.7		1.2 ± 0.7		0.1 ± 0.1		0.9 ± 0.9		0.7 ± 0.7	
Activated Charcoal		0.7 ± 0.7		2.2 ± 1.3		0.7 ± 0.7		0.1 ± 0.1		1.4 ± 1.1		0.7 ± 0.7	
Effect		*p*-value	LSD	*p*-value	LSD	*p*-value	LSD	*p*-value	LSD	*p*-value	LSD	*p*-value	LSD
DON		0.75	0.56	0.07	0.92	0.06	0.69	0.30	0.19	0.53	0.61	0.34	0.60
Additive		0.97	0.56	0.94	0.92	0.16	0.69	0.80	0.19	0.13	0.61	0.96	0.60
DON × Additive		<0.01	0.80	0.04	1.31	0.62	0.98	0.08	0.28	0.10	0.86	0.57	0.85

MD: Moderate DON level (2300 ppb in the diet); LD: Low DON level (900 ppb in the diet). MD and LD diets were given in the start (D0–14) and grower (D14–28) periods only. During the finisher period, all birds were fed a diet with negligible (57.3 ppb) DON levels. * The higher the score, the more villus damage. (a,b) Values followed by a different letter within a column differ significantly (*p* ≤ 0.05).

**Table 13 toxins-13-00170-t013:** Mean (± SD) effects of DON and activated charcoal on intestinal viscosity (cP) at D14, D28, and D35.

Treatment	DON	D14 (cP)		D28 (cP)		D35 (cP)	
No Additive	MD	16.6 ± 3.1		6.4 ± 3.8		2.2 ± 0.3	
Activated Charcoal	MD	14.6 ± 2.9		7.9 ± 2.6		2.1 ± 0.2	
No Additive	LD	9.1 ± 2.2		9.7 ± 3.7		2.2 ± 0.2	
Activated Charcoal	LD	10.5 ± 1.2		9.6 ± 6.0		2.3 ± 0.2	
	MD	15.6 ± 3.2	b	7.1 ± 3.2		2.1 ± 0.3	
	LD	9.8 ± 1.9	a	9.6 ± 4.8		2.2 ± 0.2	
No Additive		12.9 ± 4.7		8.0 ± 4.0		2.2 ± 0.3	
Activated Charcoal		12.6 ± 2.9		8.7 ± 4.5		2.2 ± 0.2	
Effect		*p*-value	LSD	*p*-value	LSD	*p*-value	LSD
DON		<0.001	1.83	0.15	3.39	0.32	0.19
Additive		0.75	1.83	0.66	3.39	0.95	0.19
DON × Additive		0.06	2.59	0.62	4.80	0.32	0.27

MD: Moderate DON level (2300 ppb in the diet); LD: Low DON level (900 ppb in the diet). MD and LD diets were given in the start (D0–14) and grower (D14–28) periods only. During the finisher period, all birds were fed a diet with negligible (57.3 ppb) DON levels. (a,b) Values followed by a different letter within a column differ significantly (*p* ≤ 0.05).

**Table 14 toxins-13-00170-t014:** Evaluated and mean levels of DON and other mycotoxins in the starter (D0–14), grower (D1–28), and finisher (D28–35) diets.

	Experimental Diets			
Mycotoxins Levels (ppb)	Moderate DON (MD)	MD + Activated Charcoal	Low DON (LD)	LD + Activated Charcoal
*Starter Diet (D0–14)*				
DON	2060	2200	878	884
DON-3-Glucoside	132	132	99	454
Enniatin B	28.2	32.3	90.4	61.8
Enniatin B1	13.1	8.5	16.0	17.3
Alternariol	10.7	-	-	3.3
Alternariol ME	-	-	-	2.4
*Grower Diet (D14-28)*				
DON	2360	2220	941	811
DON-3-Glucoside	1670	1480	851	632
Zearalenone	-	-	18.2	16.6
Ochratoxin	-	-	-	3.3
Enniatin B	36.7	41.7	58.5	60.5
Enniatin B1	8.7	10.4	15.8	16.6
Alternariol	4.2	-	3.8	2.1
Alternariol ME	2.1	2.2	2.7	-
*Finisher Diet (D14–28)*				
DON	57.3	57.3	57.3	57.3
Enniatin B	8.4	8.4	8.4	8.4
Beauvericin	6.1	6.1	6.1	6.1

## Data Availability

Data sharing not applicable.

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
