# Peer review of "Impaired Performance of Broiler Chickens Fed Diets Naturally Contaminated with Moderate Levels of Deoxynivalenol"

_toxins, 2021, doi:10.3390/toxins13020170_

Round 1
Reviewer 1 Report
please see the uploaded PDF file

Author Response
Reviewer 1
This manuscript intended to analyse / test the eects of DON (and DON+charcoal) at low, but realistic dietary concentrations on the performance of broilers. Generally, I agree with the manuscript build-up and also the results and their interpretation.
A: We acknowledge the positive comments and the suggestions/corrections made by the present reviewer. All the requests were performed and are marked with track changes.
Some points to correct or modify are listed below:
- I really understood everything in the text, with the only exception of the title. This is for me confusing. I suggest the following: Impaired Broiler Performance of Broilers Fed Diets Naturally Contaminated with Moderate Levels of Deoxynivalenol. Carry over eect for me is absolutely dierent in meaning.
A: The title of the manuscript is now “Impaired Performance of Broiler Chickens Fed Diets Naturally Contaminated with Moderate Levels of Deoxynivalenol”.
- L31-42: Please do not force the introduction of multitoxic eects once a single toxin was tested.
A: Mention to multi-contamination was removed as requested.
- Table 1. There is a number 12 in between the lines. Is this a mistake? (please see the image for indication)
A: Indeed, it is a mistake. It was removed.
- Table 6. similarly, a technical question (please see the image for indication)
A: This was also a mistake and it was removed.
- L 306: most likely because the birds were using part of their energy/nutrients to recover gut function. - what is the basis of this statement?
A: This was a suggestion. But because we do not have a robust basis for this, the statement was removed.
- L 342: please change the word decontaminant to adsorbent In summary, I did not nd critical problems or errors. After addressing my minor points I suggest acceptance of the ms. for publication in TOXINS.
A: The change was performed, and we acknowledge the pertinent suggestions and corrections from the present reviewer.
Reviewer 2 Report
The authors reported effects of low and moderately contaminated feed with DON. DON concentrations below the maximum limit of 5,000 ppb impaired broiler performance. This is an important outcome of this study and it should be further investigated in future research. Therefore, the manuscript is relevant and suitable for publication in Toxins. The manuscript is clear, well-written and the applied methodologies are of high quality. I only have minor remarks.
L13: BWG and FCR should be written in full, abbreviation is not clear until you read methods section
L51: class of ionophores: include reference or explain why this class is of most relevance
L84: include range of BW
Tables 5-8: I would include standard deviations for weight and morphometric parameters of intestines
Author Response
Comments from the reviewers:
Reviewer 2
The authors reported effects of low and moderately contaminated feed with DON. DON concentrations below the maximum limit of 5,000 ppb impaired broiler performance. This is an important outcome of this study and it should be further investigated in future research. Therefore, the manuscript is relevant and suitable for publication in Toxins. The manuscript is clear, well-written and the applied methodologies are of high quality. I only have minor remarks.
A: We acknowledge the positive comments and the suggestions made by the present reviewer. All the requests were performed and are marked with track changes.
L13: BWG and FCR should be written in full, abbreviation is not clear until you read methods section
A: Correction was done.
L51: class of ionophores: include reference or explain why this class is of most relevance
A: This class of anticoccidial fits the classical definition of an antibiotic because they have some antibacterial activity. This sentence was added in the manuscript.
L84: include range of BW
A: This information is now included: 42.4 – 43.0 g.
Tables 5-8: I would include standard deviations for weight and morphometric parameters of intestines
A: This information is now included in all the tables.